

# Seasonal variation characteristics of water quality in the Sunxi River Watershed, Three Gorges Reservoir Area

Wenning Hou, Haiyan Wang, Yonglin Zheng, Yige Wang, Dandan Yang and Hai Meng

School of Forestry, Key Laboratory for Silviculture and Conservation of Ministry of Education, Beijing Forestry University, Beijing, Haidian, China

## ABSTRACT

The seasonal change characteristics of water quality in the Sunxi River watershed, which is a typical watershed in the tail area of the Three Gorges Reservoir Area, must be studied to provide remediation ideas and specific measures for agricultural nonpoint source pollution in the reservoir area. A two-way repeated measures ANOVA was used to analyze the variation characteristics of chemical oxygen demand (COD), total nitrogen (TN), and total phosphorus (TP) concentrations in the upstream and downstream of the Sunxi River watershed in spring, summer, and autumn of 2018–2021. With data from autumn 2018 taken as an example, path analysis was applied to study the effect degree of influencing factors on TN concentrations. The two-way repeated measures ANOVA illustrated that the COD, TN, and TP concentrations in the downstream were significantly higher than those in the upstream ($P < 0.05$). In addition, the COD concentrations were the highest in summer 2019, followed by spring of 2019 and 2021, and TN and TP concentrations were higher in spring and summer. The TN and TP concentrations were comparatively lower in the autumn. The path analysis showed that electrical conductance and dissolved oxygen directly affected the TN concentrations, and water temperature mainly affected the TN concentrations via the indirect effects of electrical conductance and dissolved oxygen. The water quality of upstream Sunxi River watershed was better than that of downstream, and the water quality in autumn was better than spring and summer in 2018–2021. For water quality management and ecological restoration of the Sunxi River watershed, further attention should be paid to the water quality changes in the downstream and in spring and summer and to the impact of water temperature, electrical conductivity, and dissolved oxygen on the water quality.

## INTRODUCTION

Nonpoint source pollution has become one main factor threatening aquatic ecological environments with water eutrophication as an important manifestation. Nitrogen and phosphorus are essential nutrients for the growth and distribution of phytoplankton in aquatic ecosystems. When the concentrations of nitrogen and phosphorus reach high

Corresponding author
Haiyan Wang,
haiyanwang72@aliyun.com

levels, the risk of eutrophication occurs, leading to water quality deterioration, public drinking water safety risks, and economic losses in fisheries and other industries (*Lu et al., 2019*). Since the 21st century, the rapid growth of industrialization and population has led to water quality deterioration of major rivers and lakes in China, and most of them are facing the danger of eutrophication (*Hashim, Talib & Abustan, 2018*; *Geng et al., 2021*; *Yang et al., 2021d*). The water area with algae blooming and duckweed outbreaking will further increase if effective control measures are not implemented (*Ong & Ransangan, 2018*). As one of the largest projects of water conservancy in the world, the Three Gorges Reservoir Area is beneficial to the social and economic development of the Yangtze River watershed (*Tang et al., 2015*). However, the reservoir is facing water quality deterioration, and water quality has become a public concern due to its remarkable role in human health and ecological environment (*Huang et al., 2016*); moreover, water quality has been the focus at home and abroad since the completion of the Three Gorges Reservoir Area (*Xia et al., 2018*; *Chen et al., 2019*). With the growth of population and intensification of human activities, especially with the rapid industrial development in recent years, the amount of fertilizer application has gradually increased, resulting in the surplus of fertilizers for crop growth in farmland. Fertilizers enter the rivers through surface runoff or groundwater after rainfall or irrigation, leading to poor water quality (*Ding et al., 2013*; *Sharma & Tiwari, 2019*; *Wu et al., 2013*; *Zhu et al., 2012*).

According to the Second National Pollution Source Census Bulletin of China released in 2020, the chemical oxygen demand (COD), total nitrogen (TN), and total phosphorus (TP) accounted for 82.21%, 11.66%, and 1.21% of the water pollution discharge of the seven major basins, respectively. River water quality is not only vulnerable to anthropogenic factors, such as land use and excessive development of water resources, but also to natural factors, such as soil erosion and seasonal changes of rivers. Since the water storage in the Three Gorges Reservoir Area in 2003, eutrophication has become increasingly serious, especially in the alternation of spring and summer and summer and autumn (*Ran et al., 2013*). *Li et al. (2009)* pointed out that heavy rainfall in the Xiaojiang backwater area increased the TP concentrations in early spring, leading to the outbreak of water bloom in alternation of spring and summer in 2007. In the Hanfeng Lake watershed, the water quality in the middle and lower reaches was in moderate nutrition and mild eutrophication, which became increasingly serious in autumn and winter (*Qiao et al., 2020*). *Bu et al. (2010)* found that the water quality of the South Qinling Jinshui River gradually deteriorated from the source to downstream.

Analytical methods for water quality change with season or space mainly include Pearson correlation analysis (*Tian et al., 2022*; *Song et al., 2017*), Kriging interpolation (*Luo, Li & Wu, 2016*), cluster analysis (*Xiang et al., 2021*; *Zhang et al., 2017*), comprehensive water quality identification index (*Zhang et al., 2017*; *Zheng et al., 2021*), and Nemelo index (*Zheng et al., 2021*). Pearson correlation analysis can well reflect the correlation between water quality indicators; Kriging interpolation and cluster analysis can present intuitive results; comprehensive water quality identification index and Nemelo index can qualitatively or quantitatively evaluate water quality without being affected by extreme water quality indicators. In recent years, a one-way ANOVA has been widely used in the

spatial and temporal characteristics of water quality, indicating significant tempo-spatial differences in water quality indicators (*Tang et al., 2015*; *Ran et al., 2013*; *Rasul et al., 2017*; *Zhang et al., 2019*; *Dawson et al., 2019*; *Yang et al., 2021c*; *Zhao et al., 2016*); however, research on the Three Gorges Reservoir Area is lacking.

Some studies on water quality change with season or site in the Three Gorges Reservoir Area have been conducted with slightly different methods and indicator selections (*Zhao et al., 2016*), but only a few were conducted in the Sunxi River watershed, a typical watershed in the tail area of the Three Gorges Reservoir Area. The COD can indicate the pollution severity of water body by organic pollutants, while TN and TP can indicate the eutrophication degree of water body. Therefore, the objectives of this study were as follows: (1) to evaluate the variation of water quality from the upstream to downstream Sunxi River with the monitoring data of the COD, TN, and TP in the period of 2018–2021 using a repeated measures ANOVA; (2) to explore the factors affecting TN concentrations through path analysis using data from the autumn of 2018 as an example. By identifying the main factors that dominate the change of water quality in the Sunxi River watershed with monitoring data and distinguishing the tempo-spatial differences and the reasons, the present study provides a decision-making reference for the regulation and long-term sustainable and healthy development of the Three Gorges Reservoir Area.

## MATERIALS AND METHODS

### Study area

The Sunxi River (28°37′–29° 13′N, 106°15′–106°31′E) originates from the Jinding Mountain, Xishui County, Guizhou Province, flows through Baixi Village, Jiangjin District, Chongqing City, and finally flows into the Qijiang drainage. It belongs to the Qijiang River system with a total length of 120 km, a drainage area of 1,198 km$^2$, and an annual average flow of 19.9 m$^3$/s. The natural drop is 980 m. The vegetation type is evergreen broad-leaved forest, and the typical tree species are mainly masson pine (*Pinus massoniana* Lamb.), Chinese fir (*Cunninghamia lanceolata*), and cypress (*Cupressus funebris* Endl.). The annual average rainfall ranges from 1,000 mm to 1,450 mm, with precipitation from May to September accounting for 70%–75% of that for the whole year. A subtropical monsoon humid climate predominates, featuring an annual average temperature of 13.6 °C–18.3 °C and rain heat synchronization.

### Sample collection

Water samples were collected 3–4 km along the Sunxi River in autumn 2018, spring, summer, and autumn 2019, summer 2020 (no sampling was carried out in spring and autumn 2020 due to the outbreak of COVID-19), and spring and summer 2021, respectively. 19 samples were separately taken upstream and downstream (Fig. 1). The samples were mixed in a storage agent (1:1 concentrated sulfuric acid, adjusted to pH ≤2) and kept in a 0 °C–4 °C refrigerator for water quality determination in the lab.
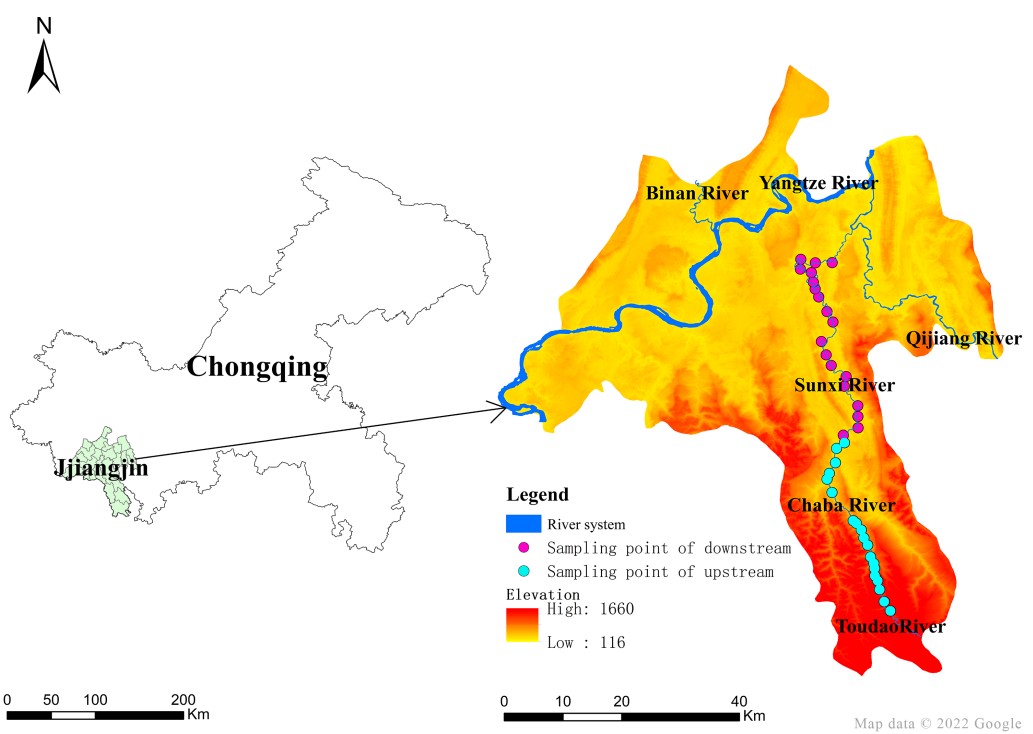

**Figure 1** **Water sampling sites in the Sunxi River watershed.**

## Water quality determination

The COD, TN, and TP were determined by dichromate, alkaline potassium persulfate digestion spectrophotometry, and ammonium molybdate spectrophotometry, respectively, in accordance with Water and Wastewater Monitoring and Analytical Methods (*Ministry of Environmental Protection of the People's Republic of China, 2002*).

## Data analysis

Herein, Microsoft Excel 2010 was used for data collation, Arc GIS 10.8 was used to map sampling sites, and SPSS 23 was applied to achieve path analysis, a two-way repeated measures ANOVA, and multiple comparisons.

The repeated measures design refers to the repeated observations of the same dependent variable of the same research subject at different times or situations to study whether significant differences exist between treatments and the interaction between factors and time (*Xu et al., 2016*). The repeated measures ANOVA requires relatively few samples, with each sample as its own control, to overcome intersample variation; thus, it is relatively simple and easy to operate. It can be applied to long-term water quality monitoring. In the present study, a two-way repeated measures ANOVA was performed to estimate the tempo-spatial variation of water quality in the Sunxi River in 2018–2021. The box–whiskers plots reveal outliers in the data, and no outliers existed after elimination by replacing the second most extreme values with the extreme outliers.

Due to the best fit of TN with water physicochemical properties in autumn 2018, the path analysis was used to analyze the effect of water physicochemical properties on TN concentrations of the Sunxi River to reflect the mechanism as an example.

# RESULTS AND ANALYSIS

## Results of a two-way repeated measures ANOVA

The variance covariance matrix of the dependent variables is equal for the interaction term of sampling sections (*i.e.,* upstream and downstream) and seasons after Mauchly's spherical hypothesis test ($P > 0.05$).

### Change characteristics of COD

The interactions of sampling sections and seasons were statistically significant for COD concentrations, with $F (3.513, 63.226) = 19.414$ and $P < 0.05$. Therefore, separate effects were tested for sampling sections and seasons.

Water quality in spring 2019 was good in accordance with the National Standard of GB3838-2002 in which a COD limit of 15 mg/L was set for grade-I lakes and reservoirs. The COD concentrations between upstream and downstream varied significantly in autumn 2018 and 2019, summer 2020, and spring 2021, but no significant differences were found in COD concentrations between upstream and downstream in other sampling seasons (Table 1, Fig. 2).

In the upstream of the watershed, seasonal factor had a significant effect on the COD concentrations ($F (3.098, 55.767) = 43.243$, $P < 0.05$). Multiple comparisons showed that COD concentrations in summer were significantly higher than those in spring and autumn ($P < 0.05$). The COD concentrations were significantly higher in spring 2019 than in spring 2021, in summer 2019 and 2021 than in summer 2020, and in autumn 2019 than in autumn 2018 (Table 1, Fig. 2).

In the downstream of the watershed, seasonal factor had a significant effect on the COD concentrations ($F (3.385, 67.692) = 42.313$, $P < 0.05$). Multiple comparisons revealed that the COD concentrations in summer 2019 were significantly higher than those in spring and autumn ($P < 0.05$). The comparison of COD concentrations in each season showed that COD concentrations were significantly lower in summer 2020 than in summer 2019 and in autumn 2019 than in autumn 2018 (Table 1, Fig. 2).

### Change characteristics of TN

The interaction of sampling sections and seasons were statistically significant for TN concentrations, with $F (2.303, 41.456) = 35.153$ and $P < 0.05$. Therefore, separate effects were tested for sampling sections and seasons.

In spring 2019, the concentrations of TN were the highest, reaching grade IV (GB3838-2002). The TN concentrations varied significantly between upstream and downstream in autumn 2018, spring 2019, summer 2020, and spring 2021, but no significant differences were found in the other sampling seasons (Table 2, Fig. 2).

In the upstream, seasonal factor had a significant effect on the TN concentrations ($F (3.297, 59.342) = 16.160$, $P < 0.05$). Multiple comparisons showed significantly higher TN

Hou et al. (2022), *PeerJ*, DOI 10.7717/peerj.14233

**Table 1  The COD concentrations of upstream and downstream Sunxi River in different sampling seasons (mg/l).**

| Group | Sample size | COD | | | | | | | P value |
|---|---|---|---|---|---|---|---|---|---|
| | | **2018** | **2019** | | | **2020** | **2021** | | |
| | | autumn | spring | summer | autumn | summer | spring | summer | |
| Upstream | 19 | $3.32 \pm 1.80^c$ | $5.63 \pm 2.03^b$ | $10.00 \pm 2.47^a$ | $6.42 \pm 1.30^b$ | $2.00 \pm 0.00^c$ | $2.00 \pm 0.00^c$ | $5.11 \pm 3.13^{bc}$ | $0.000^*$ |
| Downstream | 19 | $6.32 \pm 1.77^b$ | $6.37 \pm 2.00^b$ | $9.53 \pm 1.47^a$ | $3.53 \pm 1.65^c$ | $2.84 \pm 1.30^c$ | $8.00 \pm 1.76^{ab}$ | $6.00 \pm 2.71^{bc}$ | $0.000^*$ |
| P value between groups | | $0.001^*$ | 0.277 | 0.479 | $0.000^*$ | $0.011^*$ | $0.000^*$ | 0.352 | |

**Notes.**

The data are mean ± standard deviation.

*indicates a significant difference ($P < 0.05$).

If the same lowercase letter does not appear in the same row, it indicated significant differences in COD concentrations in different seasons ($P < 0.05$).

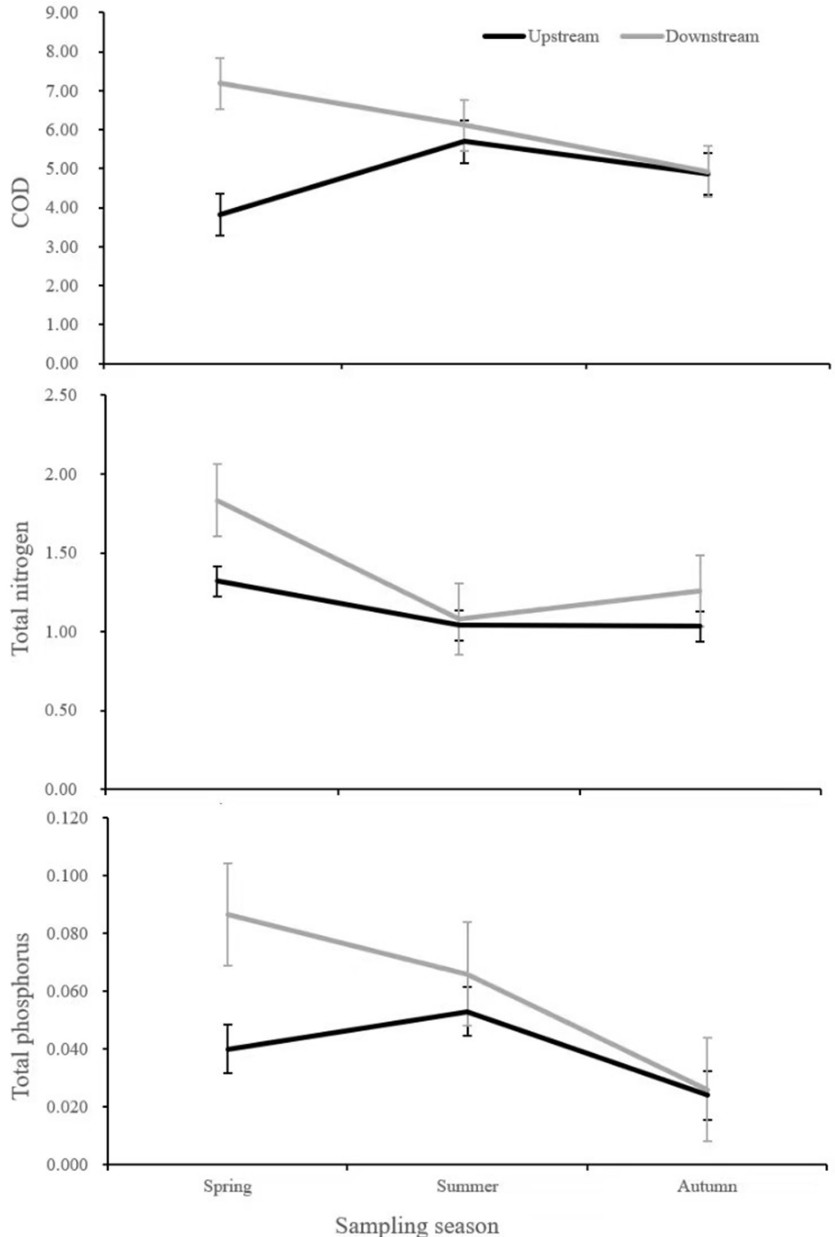

**Figure 2** The concentrations of COD, total nitrogen and total phosphorus in upstream and downstream Sunxi River in different sampling seasons (mg/l).

concentrations in spring 2019 compared with autumn 2019 and in spring 2021 compared with summer 2021 ($P < 0.05$). The comparison of TN concentrations in each season showed significantly lower TN concentrations in summer 2020 compared with 2019 (Table 2, Fig. 2).

In the downstream, seasonal factor had a significant effect on the TN concentrations ($F(2.183, 43.653) = 34.193$, $P < 0.05$). Multiple comparisons showed significantly higher TN

Hou et al. (2022), *PeerJ*, DOI 10.7717/peerj.14233

**Table 2  Total nitrogen concentrations in upstream and downstream Sunxi River in different sampling seasons (mg/l).**

| Group | Sample size | Total nitrogen | | | | | | | P value |
|---|---|---|---|---|---|---|---|---|---|
| | | **2018** | **2019** | | | **2020** | **2021** | | |
| | | autumn | spring | summer | autumn | summer | spring | summer | |
| Upstream | 19 | $0.99 \pm 0.21^{bc}$ | $1.30 \pm 0.29^{a}$ | $1.35 \pm 0.35^{a}$ | $1.08 \pm 0.15^{b}$ | $0.87 \pm 0.17^{c}$ | $1.34 \pm 0.07^{a}$ | $0.91 \pm 0.10^{bc}$ | $0.000^{*}$ |
| Downstream | 19 | $1.37 \pm 0.24^{b}$ | $1.59 \pm 0.31^{b}$ | $1.28 \pm 0.29^{bc}$ | $1.15 \pm 0.17^{c}$ | $1.01 \pm 0.09^{d}$ | $2.08 \pm 0.17^{a}$ | $0.95 \pm 0.05^{d}$ | $0.000^{*}$ |
| P value between groups | | $0.000^{*}$ | $0.009^{*}$ | 0.556 | 0.289 | $0.013^{*}$ | $0.002^{*}$ | 0.768 | |

**Notes.**
The data are mean $\pm$ standard deviation.
*indicates a significant difference ($P < 0.05$).
If the same lowercase letter does not appear in the same row, it indicated significant differences in COD concentrations in different seasons ($P < 0.05$).

concentrations in spring 2019 compared with autumn 2019 and in spring 2021 compared with summer 2021 ($P < 0.05$). The comparison of TN concentrations in each season showed significantly lower TN concentrations in spring 2019 compared with spring 2021, in summer 2020 and 2021 compared with summer 2019, and in autumn 2019 compared with 2018 (Table 2, Fig. 2).

### Change characteristics of TP

The spherical assumption was not satisfied between the upstream and downstream of the watershed. $\chi^2 = 85.005$ ($P < 0.05$); thus, the interaction effect on the TP concentrations was corrected using the Greenhouse–Geisser method, with $F$ (2.112, 38.008) = 11.762 and $P < 0.05$. Therefore, separate effects were tested for sampling sections and seasons.

The maximum concentration of TP appeared in spring 2019, reaching grade II (GB3838-2002). In spring and summer 2019 and spring 2021, TP concentrations varied significantly between upstream and downstream, but no significant differences were found in other sampling seasons (Table 3, Fig. 2).

In the upstream, seasonal factor had a significant effect on the TP concentrations ($F$ (2.629, 47.327) = 11.848, $P < 0.05$). Multiple comparisons showed significantly higher TP concentrations in spring and summer of 2019 compared with autumn 2019 ($P < 0.05$); the comparison in each season showed significantly higher TP concentrations in autumn 2018 compared with 2019 (Table 3, Fig. 2).

In the downstream, seasonal factor had a significant effect on the TP concentrations ($F$ (2.382, 42.882) = 27.721, $P < 0.05$). Multiple comparisons showed that the TP concentrations in spring and summer of 2019 were significantly higher than those in autumn ($P < 0.05$). The TP was significantly higher in spring 2019 compared with spring 2021 and in summer 2019 compared with summer 2020 and 2021, but it was significantly lower in autumn 2019 compared with autumn 2018 (Table 3, Fig. 2).

## Path analysis of TN concentrations with water physicochemical properties

With water quality measurement data in autumn 2018 taken as an example, the influence degree and effects of water physicochemical properties on TN concentrations were analyzed. Path analysis showed that water temperature, electrical conductivity (EC) and dissolved oxygen (DO) had direct and indirect effects on the TN concentrations in the water body (Table 4, Fig. 3). The absolute value of correlation coefficient was in the order of DO > EC > water temperature, where EC and water temperature showed positive effects, and DO showed a negative effect. The absolute value of direct path coefficient was shown as water temperature > DO> EC, where water temperature and DO had negative effects, and EC had a positive effect. The indirect path coefficient was also manifested as water temperature > DO> EC, but water temperature had a positive effect, followed by DO; by contrast, EC had a negative effect, indicating that water temperature affected TN concentrations mainly through the indirect effects of DO and EC.

**Table 3** Total phosphorus concentrations in upstream and downstream Sunxi River in different sampling seasons (mg/l).

| Group | Sample size | Total phosphorus | | | | | | | P value |
|---|---|---|---|---|---|---|---|---|---|
| | | **2018** | **2019** | | | **2020** | **2021** | | |
| | | autumn | spring | summer | autumn | summer | spring | summer | |
| Upstream | 19 | $0.038 \pm 0.025^{bc}$ | $0.051 \pm 0.045^{ab}$ | $0.068 \pm 0.020^{a}$ | $0.010 \pm 0.004^{d}$ | $0.045 \pm 0.010^{b}$ | $0.029 \pm 0.003^{c}$ | $0.047 \pm 0.005^{ab}$ | $0.000^{*}$ |
| Downstream | 19 | $0.040 \pm 0.017^{c}$ | $0.120 \pm 0.060^{a}$ | $0.099 \pm 0.044^{a}$ | $0.011 \pm 0.004^{d}$ | $0.055 \pm 0.022^{b}$ | $0.053 \pm 0.003^{bc}$ | $0.043 \pm 0.003^{bc}$ | $0.000^{*}$ |
| P value between groups | | 0.764 | $0.003^{*}$ | $0.007^{*}$ | 0.663 | 0.122 | $0.000^{*}$ | 0.542 | |

**Notes.**

The data are mean $\pm$ standard deviation.

*indicates a significant difference ($P < 0.05$).

If the same lowercase letter does not appear in the same row, it indicated significant differences in COD concentrations in different seasons ($P < 0.05$).

**Table 4  Path analysis of water physicochemical properties on total nitrogen concentrations in Autumn 2018.**

| Independent variable | Simple correlation coefficient with the total nitrogen | Direct path coefficient | Indirect path coefficient | | | In total |
|---|---|---|---|---|---|---|
| | | | Water temperature | EC | DO | |
| Water temperature | 0.115 | −0.717 | | 0.373 | 0.459 | 0.832 |
| EC | 0.395 | 0.523 | −0.512 | | 0.384 | −0.128 |
| DO | −0.459 | −0.655 | 0.503 | −0.307 | | 0.196 |

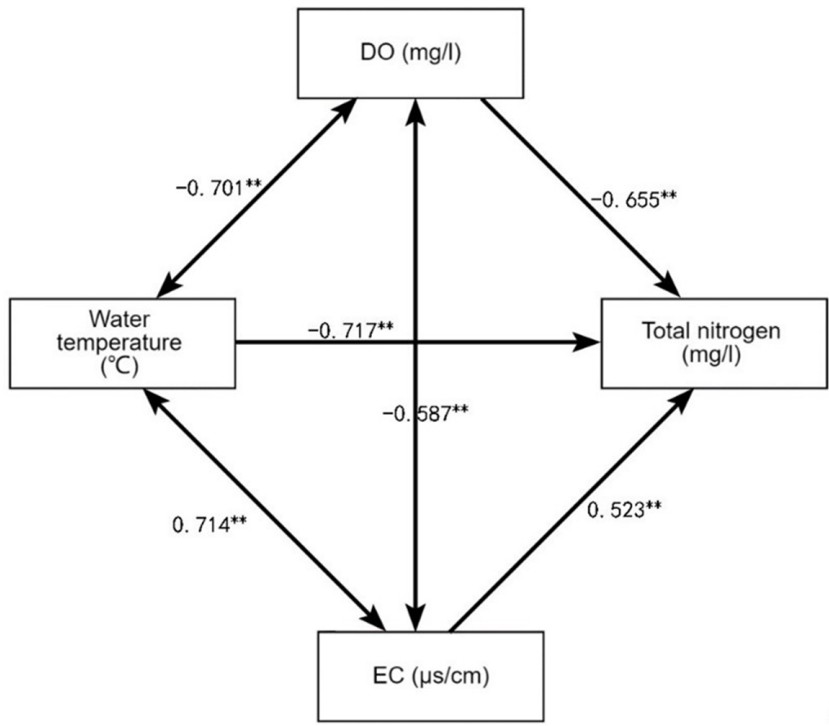

**Figure 3  The effect of water physicochemical properties on total nitrogen concentrations in Autumn 2018.**

## DISCUSSION

The two-way repeated measures ANOVA showed that the concentrations of COD, TN, and TP in the Sunxi River watershed were higher in downstream than in upstream, indicating an inferior water quality downstream; this finding is consistent with the result obtained by *Zheng et al. (2021)*. The differences in water quality between the upstream and downstream of the river watershed are related to the life of the residents and agricultural fertilization. High terrain, lush vegetation, and less human disturbance were featured in the upstream of

the Sunxi River, whereas more farmers, larger cultivated land area, and serious soil erosion were observed in the downstream. However, some opposite findings were also observed, *e.g.,* pollution was serious in the upstream and middle buffer areas because of domestic water, rural livestock, and poultry waste (*Wang et al., 2021*); the water quality upstream was inferior to downstream due to the deposition of pollutants in the wetland area of the Guanting watershed (*Yang et al., 2021b*). *Yang et al. (2022)* found that the upstream of the Jinyin Lake had many factories with large domestic sewage discharge and small and relatively closed areas, whereas large area and strong mobility downstream improved water quality. The above studies showed that the water quality of rivers was greatly affected by factories and domestic sewage, livestock and poultry manure, and farmland fertilization.

The COD concentrations were the highest in summer 2019, followed by spring 2019 and 2021. This finding is consistent with the conclusion that the water quality from September to April each year was significantly better than that from May to August (*Zhuo et al., 2017*). *Liu et al. (2020)* also remarked that the water quality from March to May and from July to August was worse than other months. The gradually increased temperature in spring and enhanced activity of organic substances in the water led to plankton bloom. In addition, the water was seriously polluted by reducing organic matter. The TN and TP concentrations were higher in spring and summer and lower in other seasons. In this study, path analysis showed a direct effect of water temperature and an indirect effect of DO on TN concentrations in the water body. The increasing DO concentration is conducive to aerobic microbial activity, which exerts an influence on the concentrations of TN by promoting oxidation and nitrification, in water (*Zhang et al., 2014*). Plankton nitrogen fixation capacity increased with the rising temperature in spring, high precipitation in summer washed nitrogen and phosphorus into water, and abundant microorganisms resulting from high temperature led to high concentrations of TN and TP in the water body (*Han et al., 2020*).

Spring is the peak period of fertilization, and nitrogen and phosphorus mainly come from the agricultural fertilizer application on both sides of the reservoir area. The TP in the watershed was also at a high level in summer because of its poor mobility in soil, which entered the watershed via surface runoff for a long time. This finding is basically in line with the conclusion of *Zhang, Li & Jiang (2020)*, who suggested that the concentrations of ammonium and nitrate were significantly higher in rainy season than dry season. Similarly, *Zhao et al. (2020)* concluded that the concentrations of TN and TP were higher in summer than in other seasons. Different geographical locations lead to differences in terrain and planting structure. *Chen et al. (2016)* pointed out that the concentrations of TN and nitrate nitrogen in the Wangjiagou watershed were significantly and positively correlated with the proportion of corn and mustard planting areas, which reached the peak after fertilization in spring and autumn. In addition, several studies have shown improved water quality in autumn compared with summer due to higher water table in autumn (*Han et al., 2020*; *Yang et al., 2012*). However, water quality in summer was also found better than other seasons. *Lv et al. (2016)* analyzed the seasonal variation characteristics of the TN and TP concentrations in the Luoshi River and believed that plankton would consume more nitrogen and phosphorus with the growth and thriving in the summer. *He et al.*

*(2021)* found high TP mostly in the dry season (September–December), whereas the wet period (May–August) was dominated with high TN due to the water stratification in the Baihua Lake of Guiyang. The results of the present study and previous ones suggest that water quality is poor in spring, good in autumn, and uncertain in summer. The conclusions varied with study areas of different terrain, land use structure, water area, flow rate, planting structure, and climate conditions. Therefore, attention should be paid to the concentrations of various indicators in summer. To evaluate the tempo-spatial characteristics of water quality comprehensively and accurately, we must analyze the change trend of distinct water layers, especially the level of phosphorus in water.

High population density and the direct discharge of domestic sewage and livestock feces into water are the main reasons for the high COD concentrations in the Sunxi River watershed. A sewage treatment system should be built by the local government to achieve centralized treatment of sewage and feces before being discharged into the river and thus ensure the water quality of the watershed accordingly. Farmers' awareness of green production should be improved because fertilization and livestock and poultry breeding are the main sources of nitrogen and phosphorus in water (*Rissman & Carpenter, 2015*; *Yang et al., 2021a*). Waste from livestock and poultry breeding should be used as organic fertilizer after compost maturity, and the amount of chemical fertilizers should be controlled to reduce the amount of nitrogen and phosphorus in surface runoff. Moreover, land use structure must be adjusted, optimal spatial layout of crops must be explored, and suitable crops must be planted in accordance with the local terrain, soil type, and soil erosion conditions. In so doing, water pollution in the watershed and the input quantity from sources by no tillage and fallow of the farmland near the watershed can be reduced. During the seasons with severe pollution of nitrogen and phosphorus, dense grass mulching (*Tu et al., 2021*) can be planted around the water area or intercepted by grass ditch (*Jiang et al., 2019*), hedge (*Zheng et al., 2020*), and penetration filter ditch (*Kong et al., 2006*). In addition, the comprehensive treatment of sewage input from the upstream should be increased to control the amount and accumulation of TN and TP in the watershed and thus reduce pollution in the downstream.

## CONCLUSIONS

Based on the water quality monitoring data in different seasons of 2018–2021 in the Sunxi River watershed, we reached the following conclusions: first, the concentrations of COD, TN, and TP were higher downstream than upstream, indicating superior water quality upstream. Second, the concentrations of COD, TN, and TP varied significantly between seasons, COD concentrations were the highest in summer and second in spring, and the concentrations of TN and TP were higher in spring and summer and lower in other seasons; lastly, EC and DO directly affected the concentrations of TN, and water temperature affected them through the indirect effect of EC and DO.

We analyzed the trends in water quality of three seasons in 2018–2021. The change of water quality has strong uncertainty, and interannual and seasonal trends with influencing factors, such as land use, pollutant discharge, and riparian environment, should be further studied.

## ACKNOWLEDGEMENTS

We thank Han Zhao and Yihong Ning for their assistance in field sampling and laboratory work.

### Funding

This research was funded by the National Key R&D Program of China (Grant No. 2017YF0505306). The funders had no role in study design, data collection and analysis, decision to publish, or preparation of the manuscript.

### Grant Disclosures

The following grant information was disclosed by the authors:
National Key R&D Program of China: 2017YF0505306.

### Competing Interests

The authors declare there are no competing interests.

### Author Contributions

- Wenning Hou conceived and designed the experiments, performed the experiments, analyzed the data, prepared figures and/or tables, and approved the final draft.
- Haiyan Wang conceived and designed the experiments, authored or reviewed drafts of the article, and approved the final draft.
- Yonglin Zheng performed the experiments, prepared figures and/or tables, and approved the final draft.
- Yige Wang analyzed the data, prepared figures and/or tables, and approved the final draft.
- Dandan Yang performed the experiments, analyzed the data, authored or reviewed drafts of the article, and approved the final draft.
- Hai Meng performed the experiments, authored or reviewed drafts of the article, and approved the final draft.

### Data Availability

The raw measurements are available as a Supplementary File.

### Supplemental Information

Supplemental information for this article can be found online at http://dx.doi.org/10.7717/peerj.14233#supplemental-information.

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
