# Peer review of "Seasonal variation characteristics of water quality in the Sunxi River Watershed, Three Gorges Reservoir Area"

_PeerJ, doi:10.7717/peerj.14233_

## Round 0.1 · original submission · Major Revisions

Dear Dr. Hou,

Kindly go through the comments of the reviewers and revise your manuscript accordingly. While uploading the revised manuscript, kindly also upload point by point response letter to the reviewers.

·

Basic reporting

The basic reporting is well done.

Experimental design

The sampling had the aim of testing spatio-temporal patterns in water quality, and testing correlations of physical variables with total nitrogen content, done over a relatively long timescale. This is a basic kind of aim, which I think the experimental design achieves.

Validity of the findings

No comment

Additional comments

line 55-56: Accounted for 82.21% of what exactly?

line 83: I think the word "individual" in this sentence needs to be changed to "sample".

line 131-132: does this just mean the furthest outlying value was deleted? If so, please state this is a concise way.

Figures: the manuscript does not present many figures, with all data shown in tables. This is fine, but many readers will find it useful to see patterns in data visually. For example, the authors emphasise the pattern about summer and spring having worse water quality, so I had the idea that maybe a summary graph could be used showing overall mean seasonal COD, nitrogen and phosphorus, to help with this emphasis.

Reviewer 2 ·

Basic reporting

The study about seasonal variation characteristics of water quality in Sunxi river watershed, Three Gorges Reservoir area, can provide some knowledge for the management of watershed to lighten non-point pollution. The region is typical both in rural area and within a typical watershed in China. However, in the presentation of manuscript, there are many defects to improve as follows:

Main aspects:
1)Your citation of Line 48-53 in text is like (Ong FS and Ransangan J. 2018), which is not correct. The correct form is (Ong and Ransangan. 2018), in which the abbreviation of First names does not occur. You need to correct all in text.
2)In structure, Line 49-53 should be arranged in front of Line 39; many methods of statistics used to analysis of data should be written in Method section such as the Greenhouse-Geisser method and the spherical assumption.
3)In method section, add the specification of a two-way repeated measures ANOVA and its significance in the study, i.e., what can it be used to explain? In detail.
4) In results, Line 201-213, it is necessary to use diagram demonstrate the correlations and fill all coefficients in block diagram obviously.
5) Writing in English in Introduction and Results are poor but Method and Discussion are slightly good. You must work hard on in Introduction and Results. In method, some information needs to add so that it is clear what you want to do.
6) In Introduction, strict assumptions is needed based on scientific gaps in the field so that you can better design the methods to validate these assumptions.

Experimental design

It needs to add the assumptions for the validation of scientific issues. Some statistical methods and their goals also present in detail here.

Validity of the findings

Validity of the findings is accepted but presentation is too poor.

Additional comments

Specific comments:
Line 39: water conservancy project-projects of water conservancy
Line 39:operation-running;
Line 40: population-Social and economic development;
Line 41: a direct and obvious threat-degradation of water quality;
Line 42: human society becomes the public;
Line 47:farmland fertilizer-the fertilizers for a growth of crops in farmland;
Line 47: delete watershed and add s behind river;
Line 56: are these ratios in all rivers, or all lakes?
Line 56-57: unclear in meaning;
Line 65-66: Correct form is “Bu et al. (2018) found that the water quality of South Qinling Jinshui River gradually deteriorated from the source to downstream”;
Line 79-82: the sentence shows a gap to the former sentence;
Line 89: were-are, you should use present tense;
Line 90: the sentence changes into: evaluate the variation of water quality from the upstream to downstream in Sunxi River;
Line 91: “in 2018-201” changes into since 2018-2021;
Line 94: “the studies on water quality variation characteristics and management” changes into “the management of water quality”;
Line 101: delete word velocity;
Line 104: “is 1000-1450” changes into “ranges from 1000 to 1450 mm”;
Line 129-132: place it to the method section, here you only present results at last;
Line 238: and thus the water was seriously polluted by reducing organic matter????

---

## Round 0.2 · Minor Revisions

Kindly consider the remaining minor improvements suggested by the reviewer.

·

Basic reporting

There were just a few changes to the language use that I thought could be made:

line 38 - I would change this to "Nonpoint source pollution has become one main factor threatening aquatic ecological environments with water eutrophication as an important manifestation"

line 46 - change "where" to "with"

line 285 - I would say that the word "controversial" here would probably be better changed to something else.

line 293 - I would delete the word "perfect" here.

Experimental design

There were some details about the use of path analysis in the experimental design that I thought could be explained more:

line 92 - please explain somewhere why for path analysis this one season in this one year were used "as an example". Why use this season/year and not another one? Why not do path analyses for all seasons in all years? If this was the only time when the necessary water temperature, EC and DO measurements were taken then please state that, and also state how the results from path analysis may differ at other seasons and years.

line 135-136 - was the data used for this path analysis from upstream or downstream, or a mix of both?

Also, it seemed confusing which seasons were sampled in which years:

line 146 - above in the methods (line 110-111) it states that water samples were taken in autumn 2018, summer 2019, all seasons 2020, and summer 2021. So I was expecting to see results from only summer in 2019, and not expecting to see this result about water quality in spring 2019. Then in the rest of the results there are many more instances where other seasons were sampled not mentioned on lines 110-111. I am not sure if I have misunderstood the methods text, or if lines 110-111 are not correct and need to be changed? Please clarify this.

Validity of the findings

no comment

Additional comments

line 64 - I do not understand why the term "spatial distribution" is included in this list, please explain about this, or maybe delete this term.

line 90 - maybe there could be added some more explanation about why, from all the different variables related to water quality, these three variables specifically were measured.

line 320 - this should be stated in the methods rather than the conclusions.

---

## Round 0.3 · accepted · Accept

Thank you very much for incorporating all the suggestions made by the reviewers in both rounds of review. Congratulations!